# Intraepithelial Lymphocyte Cytometric Pattern Is a Useful Diagnostic Tool for Coeliac Disease Diagnosis Irrespective of Degree of Mucosal Damage and Age—A Validation Cohort

**DOI:** 10.3390/nu13051684

**Published:** 2021-05-15

**Authors:** Pablo Ruiz-Ramírez, Gerard Carreras, Ingrid Fajardo, Eva Tristán, Anna Carrasco, Isabel Salvador, Yamile Zabana, Xavier Andújar, Carme Ferrer, Diana Horta, Carme Loras, Roger García-Puig, Fernando Fernández-Bañares, Maria Esteve

**Affiliations:** 1Department of Gastroenterology, Hospital Universitari Mútua Terrassa, Universitat de Barcelona, 08221 Barcelona, Spain; pruiz@mutuaterrassa.es (P.R.-R.); gcarreras@mutuaterrassa.cat (G.C.); ifajardo@mutuaterrassa.cat (I.F.); etristan@mutuaterrassa.cat (E.T.); acarrasco@mutuaterrassa.es (A.C.); isalvador@mutuaterrassa.cat (I.S.); yzabana@mutuaterrassa.es (Y.Z.); xandujar@mutuaterrassa.es (X.A.); dhorta@mutuaterrassa.cat (D.H.); cloras@mutuaterrassa.cat (C.L.); ffbanares@mutuaterrassa.es (F.F.-B.); 2Centro de Investigación Biomédica en Red de Enfermedades Hepáticas y Digestivas (CIBERehd), Instituto de Salud Carlos III, 28029 Madrid, Spain; 3Department of Pathology, Hospital Universitari Mútua Terrassa, Universitat de Barcelona, 08221 Barcelona, Spain; carmeferrer@mutuaterrassa.es; 4Department of Pediatrics, Hospital Universitari Mútua Terrassa, Universitat de Barcelona, 08221 Barcelona, Spain; rgarcia@mutuaterrassa.cat

**Keywords:** coeliac disease, flow cytometry, age, sex, lesion grade, intraepithelial lymphocytes TCRγδ^+^

## Abstract

Introduction: The study of intraepithelial lymphocytes (IEL) by flow cytometry is a useful tool in the diagnosis of coeliac disease (CD). Previous data showed that an increase in %TCRγδ^+^ and decrease of %CD3^−^ IEL constitute a typical CD cytometric pattern with a specificity of 100%. However, there are no data regarding whether there are differences in the %TCRγδ^+^ related to sex, age, titers of serology, and degree of histological lesion. Study aims: To confirm the high diagnostic accuracy of the coeliac cytometric patterns. To determine if there are differences between sex, age, serology titers, and histological lesion grade. Results: We selected all patients who fulfilled “4 of 5” rule for CD diagnosis (*n* = 169). There were no differences in %TCRγδ^+^ between sexes (*p* = 0.909), age groups (*p* = 0.986), serology titers (*p* = 0.53) and histological lesion grades (*p* = 0.41). The diagnostic accuracy of complete CD cytometric pattern was: specificity 100%, sensitivity 82%, PPV 100%, NPV 47%. Conclusion: We confirmed, in a validation cohort, the high diagnostic accuracy of complete CD pattern irrespective of sex, age, serology titers, and grade of mucosal lesion.

## 1. Introduction

The diagnosis of coeliac disease (CD) is based on several criteria including positive serology, a spectrum of duodenal damage and clinical symptoms and/or risk conditions, and response to a gluten-free diet (GFD) in patients bearing the HLA-DQ2 or DQ8 genotypes. When some of these criteria are lacking, especially when serology is negative or the duodenal atrophy is not complete, the CD diagnosis is a challenge [1]. In these difficult situations, the study of duodenal intraepithelial lymphocytes (IEL) by flow cytometry is a useful tool for CD diagnosis. It has been shown to be of value in the diagnosis of CD with atrophy [2,3,4,5] and refractory CD [6,7]. An increase in %CD3^+^ TCRγδ^+^ IEL (%TCRγδ^+^) with a decrease in %CD3^−^ IEL (%CD3^−^) has been described as the typical pattern of CD [8].

The diagnosis of CD in the case of mild histological lesions (Marsh 1) can be difficult due to low sensitivity of serology and low specificity of the lymphocytic enteritis [9,10]. However, the diagnosis of Marsh 1 patients with CD is important because they present with similar clinical symptoms to patients with atrophy that reverse with a gluten-free diet (GFD) [11,12]. Previous ESPGHAN guidelines suggest that both an increase in %TCRγδ^+^ count assessed by immunohistochemical analysis of biopsies and the presence of IgA anti-tissue transglutaminase (anti-TG2) deposits increase the chances of a diagnosis of CD [7].

The increase of %TCRγδ^+^ has occasionally been found in some other conditions such as cow’s milk intolerance, food allergy, cryptosporidiosis, Giardiasis, Sjögren syndrome, Olmesartan enteropathy, and IgA deficiency. Nevertheless, the increase in %TCRγδ^+^ in these diseases tends to be mild and transient [13]. CD is the only disease in which %TCRγδ^+^ has been found to be systematically and permanently increased, even in patients following a GFD. The concomitant decrease in %CD3^−^ provides increased specificity for the diagnosis [14]. Therefore, this particular cytometric pattern may be used to confirm the CD diagnosis in patients that had already started a GFD before the diagnosis confirmation.

A previous study by our group demonstrated good diagnostic accuracy (sensitivity 85%, specificity 100%, PPV 100%, and NPV 72%) for the typical CD cytometric pattern (increased %TCRγδ^+^ and decreased %CD3^−^) in the diagnosis of CD in patients with positive serology, both Marsh 1 and Marsh 3 [8]. However, these findings should be confirmed with a larger validation cohort.

Another important issue is learning whether the cut-off values established for %TCRγδ^+^ and %CD3^−^ reveal a cytometric CD pattern influenced by age, sex, and degree of histological lesion. In this sense, the information is very limited, but it has been suggested that γδ^+^ IEL decreases with age [6].

The aims of our study were to determine: (1) whether there are differences in the percentage of TCRγδ^+^ IEL in CD patients related to sex, age, degree of histological lesion, levels of serology; and (2) the diagnostic accuracy in a large validation cohort of the typical cytometric CD pattern and of the increase in %TCRγδ^+^ IEL, without the simultaneous decrease in %CD3^−^.

## 2. Materials and Methods

### 2.1. Patients and Controls

For the period of January 2013 to December 2019, we prospectively included all patients who fulfilled CD diagnostic criteria based on the rule of ‘4 of 5’ proposed by Catassi and Fasano [1]: typical symptoms of CD, positivity of serum coeliac disease IgA class autoantibodies, HLA-DQ2 or DQ8 genotypes, coeliac enteropathy at the small intestinal biopsy, response to the GFD (at least 4 of 5 diagnostic criteria or 3 of 4 if the HLA Genotype is not performed). The control group consisted of patients referred to the gastroenterology department for endoscopic assessment including duodenal biopsy (histopathology and flow cytometry) because they had digestive symptoms or/and anemia. Digestive symptoms were defined by the chronic or intermittent presence of either diarrhea, dyspepsia, bloating, and/or abdominal pain. Controls were consecutively included based on the following criteria to rule out CD: (1) negative coeliac serology, (2) negative HLA-DQ2.5 and HLA-DQ8, and (3) normal duodenal biopsy. We excluded patients with intake of NSAIDs and Olmesartan, and patients with Crohn’s disease, autoimmune disease-associated enteropathy, collagenous sprue associated with collagenous colitis, lymphocytic enteritis due to intestinal parasitosis or *Helicobacter pylori*, and selective IgA deficiency. All CD patients and controls were recorded in a prospective maintained registry.

We performed coeliac serology, HLA genotyping, and duodenal biopsy assessment for histopathology and lymphocyte subpopulations by flow cytometry in all patients and controls.

### 2.2. Coeliac Serology

Serum IgA-tissue transglutaminase antibody (anti-TG2) and IgA titers were analyzed in serum using a quantitative automated ELISA detection kit (Elia CelikeyTM, Phadia AB, Freiburg, Germany) with recombinant human TG2 as antigen. A value of anti-TG2 ≥8 U/mL was established as the cut-off for normality [15]. Values between 2–8 U/mL were considered as a positive CD serology when titers higher than 1/40 of serum IgA anti-endomisal antibodies (EmA) were also found [16].

### 2.3. HLA Genotyping

We used a commercial reverse hybridization kit for the determination of CD heterodimers in the HLA genotyping (HLA-DQ2 [A1*0501/0505, B1*0201/*0202], HLA-DQ8 [A1*0301, B1*0301]). HLA-DQ2.5 haplotype is present in 24% of healthy controls and 90% of CD patients in our area [17]. In this study, we considered a positive coeliac genetic when the presence of HLA-DQ2.5, HLA-DQ8 or both was detected [18]. Considering the low frequency of the presence of HLA-DQ2.2 or only one allele of HLA-DQ2 haplotype in CD patients, either DQA1*05 or DQB1*02, the presence of these alleles was allowed in control individuals.

### 2.4. Duodenal Biopsy Assessment for Histopathology

Four endoscopic biopsies were taken from the second-third portion of the duodenum and one from the duodenal bulb, and these were processed using hematoxylin/eosin staining and CD3 immunophenotyping. Marsh 1 lesion (lymphocytic enteritis) was defined by 25 or more IEL per 100 epithelial nuclei along with normal villous architecture. Two endoscopic biopsies from antrum were also taken to investigate *Helicobacter pylori* infection in all patients. The lymphocyte count was performed as previously described [19,20]. Control group patients were separated into two subgroups according to the percentage of IEL (≥ or < than 18%) since some authors have suggested that a lower cut-off point should be established to redefine lymphocytic enteritis [21].

### 2.5. Duodenal Biopsy Assessment by Flow Cytometry

We performed IEL flow cytometry in all patients and controls by taking a duodenal sample from the second-third portion of the duodenum. The sample was obtained using a 2.8 mm biopsy forceps (Radial Jaw 4, Boston Scientific^®^, Marlborough, MA, USA), and immediately processed as previously described [4,8,12].

Briefly, IELs were isolated by gentle rotation in an orbital shaker at 12 rpm for 90 min in a solution of 1 mM DTT and 1 mM EDTA in 10%FBS HBSS, at room temperature. After two washes with HBSS (10 min, 300 g) IEL mixture was immediately stained for 15 min with the antibody mix described in Table 1. Viability (>90%) was assessed by trypan blue exclusion in Neubauer chamber. IELs were acquired in a four-colour FACSCalibur and analyzed with the Cell-Quest Software (BD Biosciences). PMT voltages and compensation values were manually adjusted using single stained samples. Live IELs were gated on CD45 and low scatter basis, and intraepithelial origin was confirmed with CD103 staining. (>90%).

Four cytometric patterns were described using the TCRγδ^+^ and CD3^−^ IEL percentages: First cytometric pattern was defined by an increase of %TCRγδ^+^ (>8.5%) and a decrease in %CD3^−^ (<10%) and was labeled as a complete CD IEL flow cytometric pattern (complete FCP). A second cytometric pattern was defined by an isolated increase in %TCRγδ^+^ and was labeled an incomplete CD IEL flow cytometric pattern (Incomplete FCP). The third and fourth patterns were defined as non-CD patterns: one of them was defined by an isolated decrease in %CD3^−^ and the other, labeled normal cytometric pattern, was defined by a TCRγδ^+^ ≤ 8.5% plus CD3^−^ > 10%. This corresponds to the normal cut-off established in our laboratory [8,12]. Gating strategy and the four patterns are illustrated in Figure 1.

### 2.6. Statistical Analysis

The results were expressed as mean ± SEM or median (Interquartile range, IQR) or as proportions (and their 95% confidence interval -CI- when appropriate). In order to assess the relationship between age and %TCRγδ^+^ values, the age was classified in 7 groups (0–10 years, 11–20 years, 21–30 years, 31–40 years, 41–50 years, 51–60 years, ≥61 years). To compare %TCRγδ^+^ related to anti-TG2 serum titers, three groups were stablished: patients with anti-TG2 titers ≥ 30 U/mL, between 8–30 U/mL and between 2–8 U/mL plus EmA higher than 1/40. We used a student *t* test or ANOVA test for comparing %TCRγδ^+^ cells related to sex, degree of histological damage, and serology. The non-parametric counterpart (Kruskall–Wallis test) was used to compare the different groups of age because they do not follow a normal distribution assessed by a Kolmogorov–Smirnov test. In addition, we performed a Bonferroni test to assess differences among groups. Sensitivity, specificity, negative predictive value (NPV), and positive predictive value (PPV) for the complete CD pattern and the isolated increase in %TCRγδ^+^ were calculated using 2 × 2 tables. Statistical analysis was performed using the SPSS for Windows statistical package (SPSS Inc., Chicago, IL, USA).

### 2.7. Ethical Statements

The study was conducted according to the guidelines of the Declaration of Helsinki. All participants (or their parents in the case of patients less than 16 years old) provided written informed consent. This study is part of a larger registry that prospectively collects all patients who need to be evaluated to rule out CD. This registry was approved by the Ethics Committee of the Hospital Universitari Mútua Terrassa at the start of the registry in 2010 (Code: EO/1011; date: 25 March 2010). Researchers guaranteed strict measures for preserving patient confidentiality.

## 3. Results

We included 169 patients who fulfilled CD diagnostic criteria (119 women; mean age 18.8 ± 1.5 years, range 1–83 years). One hundred forty-four patients showed villous atrophy (Marsh 3a type, *n* = 21; and 3b-c type, *n* = 123). Twenty-five patients showed architecturally normal small intestinal mucosa with an increase in IEL counts (Marsh type 1 lesion, mean age 36.00 ± 4.48 years, range 4–83 years).

In Table 2 and Figure 2, the percentages of TCRγδ^+^ in groups of different degrees of histological lesion, sex, age, and anti-TG2 serum titers are shown. No differences were found relative to any of these variables.

The control group included 49 subjects (35 women; median age 40.00 (25.00–51.50) years, range 1–67 years). Median value of IEL% was 16.70 (11.50–20.00). Subjects in the control group with IEL count <18% (*n* = 27; 20 women, median age 46.00 (35.00–53.00) years, range 1–67 years) had a median %TCRγδ^+^ of 3.36 (2.63–5.64), whereas controls with an IEL count ranging from 18 to 25% (*n* = 22) had a median %TCRγδ^+^ of 3.53 (2.59–7.89). Clinical characteristics of the control group are detailed in Table 3.

In Table 4, the four different FCPs found in CD patients and controls are shown. In CD patients, these patterns are provided depending on the degree of histological damage and in controls taking into account whether they had a percentage of IEL <18% or <25%. The majority of patients in the control group had a normal cytometric pattern; only eight of them showed abnormal patterns. Three of them showed an incomplete FCP (isolated increase of %TCRγδ^+^ (>8.5%) and the other five showed a selective decrease of %CD3^−^ (non-coeliac pattern). It must be noted that all three patients with the incomplete FCP had an IEL count between 18–25% and there were no controls showing a complete CD pattern. Therefore, none of the controls with IEL count <18% had a coeliac related FCP.

Among CD patients with atrophy (*n* = 144), 83% had a complete FCP, whereas 13.8% (*n* = 20) had an incomplete FCP and 2.8% (*n* = 4) a normal pattern. A similar picture was found for Marsh 1 CD patients (*n* = 25), with 76% having a complete FCP (76%), 16% (*n* = 4) an incomplete FCP, and 8% (*n* = 2) a selective decrease in %CD3^−^. Thus, more than 90% of CD patients irrespective of the degree of mucosal damage showed CD related FCP.

Sensitivity, specificity, NPV, and PPV of complete FCP and of the increase of %TCRγδ^+^ were calculated considering both control subjects with IEL under 18% (*n* = 27) (Table 5) and all patients in the control group with IEL under 25% (*n* = 49) (Table 6). We found that complete FCP had an 82% sensitivity, 100% specificity, and 100% PPV irrespective of the criteria of IEL normality (below 18% or 25%). By contrast, the more restrictive criteria of IEL normality (<18%) should be adopted only if increased values of %TCRγδ^+^ are used as a diagnostic tool, reaching in this case an accuracy close to that obtained with the complete FCP. The largest differences in diagnostic accuracy between the two coeliac FCPs, depending on what we consider normal duodenal mucosa (IEL count < 18% or <25%), were in the NPV. In this sense, the highest probability of not having a CD corresponded to individuals having an IEL count < 25% (non-restrictive criteria of normality) and not having an increased %TCRγδ^+^ (NPV 88%).

## 4. Discussion

The complete IEL cytometric pattern of CD, characterized by an increase of %TCRγδ^+^ and a concomitant decrease in %CD3^−^, has been proposed as a in complementary diagnostic tool to reinforce CD diagnosis in doubtful cases, especially when serology is negative [8,22]. This situation may occur in 30% of patients with atrophy and in more than 70% of patients with lymphocytic enteritis or Marsh type 1 CD [23,24].

The most frequent etiology of seronegative duodenal atrophy in Western countries is CD and the percentage increases in patients with positive HLA-DQ2/DQ8 [25]. The diagnosis of CD in cases of lymphocytic enteritis (Marsh 1 lesion) is more challenging since the lesion is much more unspecific than atrophy and other possible etiologies have been proposed [25,26]. As mentioned, only a small percentage of these patients will show a positive coeliac serology and only some patients will progress to villous atrophy after a gluten challenge of eight weeks [27].

The CD diagnosis in seronegative patients is based on the clinical and histological response to a GFD in patients with signs and symptoms of the coeliac spectrum in the presence of a positive HLA-DQ2/-DQ8. This means that the diagnosis of CD is time-consuming and remains uncertain until the effect of a GFD is assessed. In addition, gluten challenge is not well accepted by patients due to the discomfort caused. Nevertheless, it must be considered that this evaluation is sometimes difficult because CD clinical symptoms are quite unspecific and lymphocytic enteritis in non-CD patients may resolve spontaneously [19].

In the present validation cohort, we have confirmed that assessment of the complete FCP is a useful diagnostic tool for CD diagnosis, with a high diagnostic accuracy (82% sensitivity and 100% specificity). In addition, TCRγδ^+^ IEL subpopulation, which is the main parameter of coeliac lymphogram, is not influenced by age, sex, or the degree of histological damage. Hence, the IEL study through flow cytometry for CD diagnosis can be applied in any situation regardless of the clinical characteristics of the patient. This study also confirms that the normality cut-off previously established for %TCRγδ^+^ [8] is appropriate in patients bearing the complete coeliac FCP, including elderly patients. However, taking into account that the number of CD patients and controls older than 61 was very small, information focused on this population group is awaited.

Our study was performed in patients with positive serology, to ensure the diagnosis of CD, but it is conceivable that the characteristic behavior of duodenal intraepithelial subpopulations is also maintained in patients with negative serology. In fact, %TCRγδ^+^ values are not influenced by the levels of serum anti-TG2 titters. Moreover, the results of other studies by our group, showing very high response rates to a GFD in patients with enteropathy of the CD spectrum, negative serology, and coeliac cytometric pattern, lend support to this hypothesis [12,28].

A limitation of studies assessing diagnostic tools in CD is selection of the control group, and this feature of our study merits special mention. The ideal controls should be individuals of the general population who are completely healthy, without digestive symptoms and with negative genetic predisposition and serology. To our knowledge, a study with this type of ‘perfect’ control group has never been performed. In fact, the cut-off of 18 IEL considered ‘normal’ in the duodenal mucosa was established in subjects in whom the duodenal mucosa was microscopically assessed due to digestive symptoms [21]. The CD was ruled out with negative serology and negative HLA-DQ2/DQ8. In our study, we also used the same criteria for control group recruitment, excluding all the individuals with a positive HLA-DQ2.5 and DQ8. The recruitment of these controls was consequently very slow because the percentage of individuals in the general population having either HLA-DQ2 or DQ8 exceeds 60% in our area [17], but this makes the diagnosis of CD almost impossible.

Eight subjects in the whole control group had an abnormal FCP. Three of them had an incomplete FCP and the remaining 5 a selective decrease of %CD3^−^. By contrast, none of the controls with IEL count < 18% had a coeliac FCP and only two of them had a selective decrease of %CD3^−^, highlighting how this value should be considered the normal cut-off for histopathological analysis. Consequently, we noted a slight decrease in the diagnostic accuracy when we considered the sub-group that presented an IEL count between 18–25% as controls. All these findings are objective data to redefine the cut-off point <18% for considering duodenal mucosa as normal. Also, it is demonstrated that complete FCP is more accurate than an incomplete CD pattern.

In conclusion, the established normality cut-off for %TCRγδ^+^ (>8.5%) in IEL flow cytometry study for diagnosis of CD is valid for all age, sex, and histological lesion grade groups. Moreover, we have confirmed the high diagnostic accuracy of the increase in %TCRγδ^+^ and the complete FCP for CD diagnosis in a large validation cohort.

## Figures and Tables

**Figure 1 nutrients-13-01684-f001:**
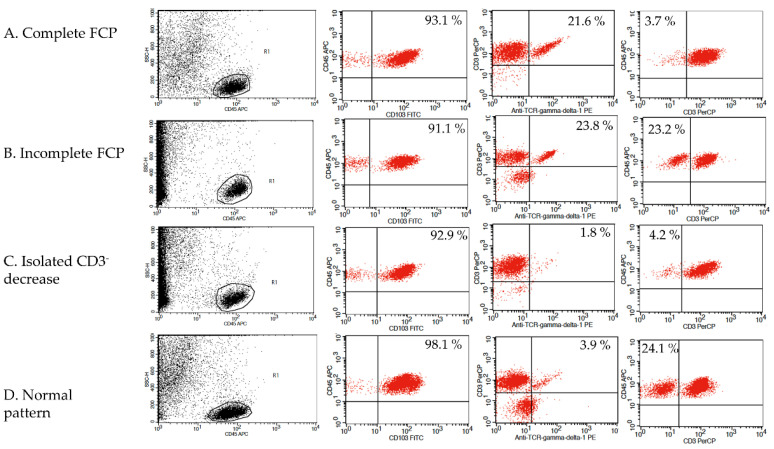
Gating strategy and the four patterns cytometric patterns. Complete and incomplete flow cytometric patterns (FCP) are CD related patterns.

**Figure 2 nutrients-13-01684-f002:**
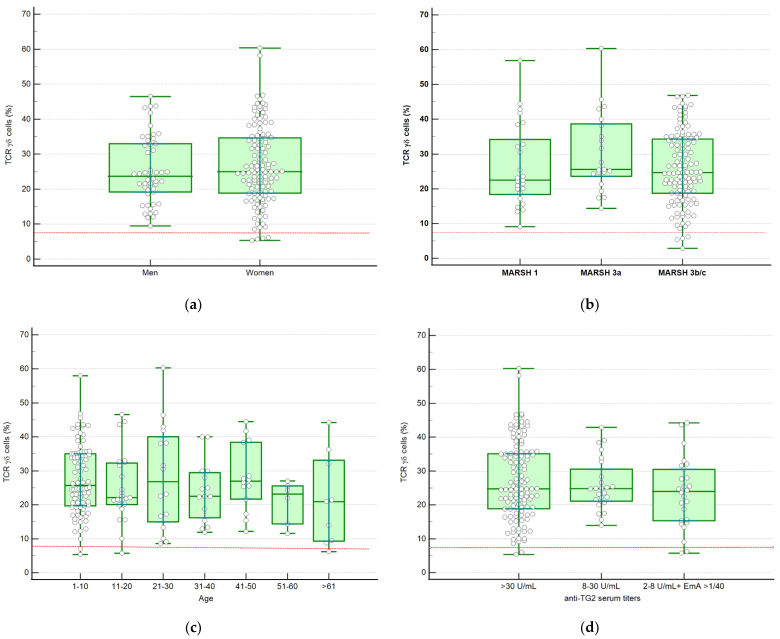
Scatter plot and box-whisker showing the distribution of patients according to sex (**a**), degree of histological lesion (**b**), age (**c**), and anti-TG2 serum titers (**d**). Box-plot rectangle spans the interquartile range, the segment inside the rectangle shows median whereas the whiskers above and below plot, the maximum and the minimum. The dotted red line represents the stablished TCRγδ^+^ cut-off (>8.5%).

**Table 1 nutrients-13-01684-t001:** Antibodies used for flow cytometry staining

Laser	Fluorochrome	Cell Marker	Antibody Clone	Supplier	Reference	Dilution
488	PerCP	CD3	SK7	BD ^1^	345,766	2.5:100
	FITC	CD103	Ber-ACT8	BD	333,155	2.5:100
633	PE	TCRγδ	11F2	BD	333,141	2.5:100
	APC	CD45	2D1	BD	340,910	1.5:100

^1^ BD: BD-Biosciences.

**Table 2 nutrients-13-01684-t002:** Comparison of %TCRγδ^+^ between different groups of sex, age, and degree of histological lesion

	Variable	Median %TCRγδ^+^ (IQR)	*p*
Sex	Male (*n* = 50)	23.70 (18.08–34.00)	0.909
	Female (*n* = 119)	25.40 (18.78–35.31)	
Histology	Marsh 1 (*n* = 25)	22.51 (16.40–35.62)	0.41
	Marsh 3a (*n* = 21)	25.60 (22.85–39.13)	
	Marsh 3b-c (*n* = 123)	24.70 (18.73–34.48)	
Age	0–10 (*n* = 86)	25.03 (19.32–35.04)	0.79
	11–20 (*n* = 23)	22.13 (20.08–32.31)	
	21–30 (*n* = 16)	26.82 (14.98–40.07)	
	31–40 (*n* = 15)	22.53 (16.19–36.38)	
	41–50 (*n* = 14)	26.98 (21.69–38.44)	
	51–60 (*n* = 6)	23.17 (14.38–25.59)	
	≥61 (*n* = 9)	21.47 (12.00–38.28)	
Serology	anti-TG2 ≥30 U/mL (*n* = 119)	24.75 (19.20–35.31)	0.53
	anti-TG2 8–30 U/mL (*n* = 24)	24.81 (20.90–33.60)	
	anti-TG2 2–8 U/mL + EmA > 1/40 (*n* = 26)	23.98 (15.18–31.70)	

**Table 3 nutrients-13-01684-t003:** Clinical characteristics of the control group.

Age (years) *	40.00 (25.00–51.50)
Sex (% women)	71.4%
Clinical symptoms ^1^	
Diarrhea	19 (36%)
Bloating	10 (20%)
Dyspepsia	10 (20%)
Abdominal pain	4 (8%)
Anaemia	4 (8%)
Autoimmune disease	4 (8%)
HLA Genotyping	
HLA-DQ2.2	16 (32%)
HLA-DQA1 * 05	14 (29%)
HLA-DQB1 * 02	9 (19%)
Without risk alleles	10 (20%)
IEL count (%) *	16.70 (11.50–20.00)
CD3^+^ TCRγδ^+^ IEL (%) *	3.40 (2.63–5.78)
CD3^−^ IEL (%) *	21.03 (13.79–30.55)
Final diagnosis	
Irritable bowel syndrome	25 (51%)
Fructose malabsorption	8 (17%)
Gastroesophageal reflux disease	6 (12%)
Lactose malabsorption	3 (6%)
Non-coeliac gluten sensitivity	2 (4%)
Autoimmune pancreatitis	1 (2%)
Chronic pancreatitis and exocrine pancreatic insufficiency	1 (2%)
Factitious diarrhea	1 (2%)
Esophageal dysmotility due to systemic sclerosis	1 (2%)
Control biopsy after Helicobacter pylori eradication	1 (2%)

^1^ If patients reported more than one symptom, the predominant one was selected. * Median (IQR).

**Table 4 nutrients-13-01684-t004:** Cytometric patterns in CD patients and control patients.

	CD Patients *n* = 169			Controls (*n* = 49)	
	Marsh 1 (*n*= 25)	Marsh 3a (*n*= 21)	Marsh 3b-c (*n*= 123)	IEL < 18 (*n* = 27)	IEL < 25 (*n* = 49)
Complete FCP	19	19	101	0	0
Incomplete FCP: Isolated increase of %TCRγδ^+^ IEL	4	2	18	0	3
Isolated decrease of % CD3^−^	2	0	0	2	5
Increase of %TCRγδ^+^ IEL ^1^	23	21	119	0	3
Normal pattern	0	0	4	25	41

FCP = Flow cytometric pattern. Complete coeliac FCP: CD3^+^ TCRγδ^+^ IEL > 8.5% and CD3^−^ < 10%. Incomplete coeliac FCP: isolated increase of CD3^+^ TCRγδ^+^ IEL > 8.5%. ^1^ Total number of patients with increase in %TCRγδ^+^ (complete + incomplete FCP).

**Table 5 nutrients-13-01684-t005:** Accuracy of coeliac cytometric pattern for the diagnosis of coeliac disease. Control group subjects with IEL count < 18% (*n* = 27).

	Sensitivity % (95% CI)	Specificity % (95% CI)	PPV % (95% CI)	NPV % (95% CI)
Complete FCP	82 (75–88)	100 (84–100)	100 (82–100)	47 (34–61)
Increase of %TCRγδ^+^ IEL ^1^	96 (92–98)	100 (84–100)	100 (97–100)	81 (64–92)

FCP = Flow cytometric pattern. Complete coeliac FCP: TCR CD3^+^ γδ^+^ IEL > 8.5% and CD3^−^ < 10%. ^1^ Total number of patients with increase in %TCRγδ^+^ (complete + incomplete FCP).

**Table 6 nutrients-13-01684-t006:** Accuracy of the coeliac cytometric pattern for the diagnosis of coeliac disease. Control group subjects under 25% IEL (*n* = 49).

	Sensitivity % (95% CI)	Specificity % (95% CI)	PPV % (95% CI)	NPV % (95% CI)
Complete FCP	82 (75–88)	100 (91–100)	100 (97–100)	62 (50–73)
Increase of %TCRγδ^+^ IEL ^1^	96 (92–98)	93 (82–98)	98 (93–99)	88 (76–95)

FCP = Flow cytometric pattern. Complete coeliac FCP: TCR CD3^+^ γδ^+^ IEL > 8.5% and CD3^−^ < 10%. ^1^ Total number of patients with increase in %TCRγδ^+^ (complete + incomplete FCP).

## Data Availability

File1.sav, database of the study (Appendix A).

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
