# Peer review of "Intraepithelial Lymphocyte Cytometric Pattern Is a Useful Diagnostic Tool for Coeliac Disease Diagnosis Irrespective of Degree of Mucosal Damage and Age—A Validation Cohort"

_nutrients, 2021, doi:10.3390/nu13051684_

Round 1
Reviewer 1 Report
Ruiz-Ramírez et al. studied the cytometric pattern of TCRγδ + and CD3- IEL for celiac disease diagnosis in a considerable group of persons varying in age, sex and histological lesion. The manuscript is mostly well written, and the authors reported some interesting results, particularly regarding the diagnosis of celiac disease in cases of lymphocytic enteritis (Marsh 1 lesion). I think the manuscript can be accepted in its present form.
No lines!
mean (SEM)?
Author Response
As a suggestion of one reviewer, we applied the Kolmoroff Smirnov test to confirm the normal distribution and we realised that the age groups didn’t follow a normal distribution. So, we applied a non-parametric test (Kruskall Wallis test) to compare TCR gd+ IELs% in different age groups. The rest of the groups we compared (sex, degree of histological damage and serology) followed a normal distribution. Therefore, parametric tests were applied (Student t test or ANOVA). Section Results, Table 2
In order to show the results in a homogeneous way, we expressed all the values (those following a normal distribution or not) as a median and IQR (interquartile range) , replacing mean +/- standard error of the mean (SEM)
Thank you
Reviewer 2 Report
The authors in their previous study describe the pattern of increased TCR gd+ IELs and decrease in CD3-IEL as a diagnostic tool in Celiac disease. In the current manuscript the authors have re-validated their earlier results (as well as similar findings by other groups) in an expanded cohort with control groups.
- The authors need to explain in better detail how and in what circumstances their findings could be used for CD diagnosis.
- Table 1 – The mean TCR gd+ for the females has large error value 27.3 +/- 41?
- It is very unusual that the authors have got a very homogenous distribution of TCRgd+ IEL in almost all the patients!
- Why have the authors used Mean values in their statistics, why not Median? Were there any outliers in their data?
- A scatter plot with individual %s will add transparency to the data.
- Did the authors find differences or associations in the TCR gd+ IEL or CD3- IELs when patients are grouped according to levels of anti-TG2 antibodies?
- Although the authors have referenced their own work for Flow cytometry methodology, it lacks details such as use of live/dead stain or Fc block to prevent false positive stainings, dilutions of the antibodies etc.
- The authors need to be indicated clearly where (%) or counts are used.
Reviewer 3 Report
Article lacks very important methodological data. Authors claim that IEL gdT >8.5% are relatively sensivity and specific marker of coeliac disease, but in the sime time, they fail to provide the readers with full description of methodology that would enable everyone to fully repeat the study. Those details are scattered across several publications. Moreover, there are no dot-plots which show the quality of flow cytometry AND gating strategy. Mere percentages are not enough. Finally, I have ethical concerns regarding the data - from my perspective it looks like salami slicing - part of the data has already been published. All those things have to be improved or clearly mentioned in the text.
For the clarity, my main points:
1. Sample handling, staining protocol, antibodies and equipment used have to be clearly indicated. Please follow the rules of the MiFlowCyt (10.1002/cyto.a.20623).
2. Please present examples of cytometry dot-plots with gating strategy and comparison between control and CD.
3. "In addition, we performeda Bonferroni test to assess where thedifferences among groupswere" - what does this mean? I assume you mean the Bonferroni correction for multiple comparisons.
4. What is the novelty in comparison to already published data from authors group - 10.3390/nu11051050 10.1111/apt.15663 10.3390/nu11091992
Round 2
Reviewer 3 Report
1. "PMT and 159compensation values were manually adjusted using single stained samples." - PMT voltages
2. "1mM DTT and 1mM EDTA " - no enzymes?
3. "degree of hystological damage and serology" - histological?
Author Response
Comments 1, 3:
Modified in the main text, thank you. (Pages 4 and 5, section: Materials and Methods)
Comment 2:
No, enzymes are used for isolation of lamina propria lymphocytes, which is not in the scope of this article. For epithelium and intraepithelial lymphocytes isolation, calcium chelation methods (i.e. DTT and EDTA treatment) are used. This is described in PMID: 9579607 and reviewed in PMID: 20833175.